# Mother-daughter asymmetry of pH underlies aging and rejuvenation in yeast

**Kiersten A Henderson, Adam L Hughes[†], Daniel E Gottschling***

Division of Basic Sciences, Fred Hutchinson Cancer Research Center, Seattle, United States

**Abstract** Replicative aging in yeast is asymmetric–mother cells age but their daughter cells are rejuvenated. Here we identify an asymmetry in pH between mother and daughter cells that underlies aging and rejuvenation. Cytosolic pH increases in aging mother cells, but is more acidic in daughter cells. This is due to the asymmetric distribution of the major regulator of cytosolic pH, the plasma membrane proton ATPase (Pma1). Pma1 accumulates in aging mother cells, but is largely absent from nascent daughter cells. We previously found that acidity of the vacuole declines in aging mother cells and limits lifespan, but that daughter cell vacuoles re-acidify. We find that Pma1 activity antagonizes mother cell vacuole acidity by reducing cytosolic protons. However, the inherent asymmetry of Pma1 increases cytosolic proton availability in daughter cells and facilitates vacuole re-acidification and rejuvenation.

## Main text

During replicative aging in budding yeast, mother cells produce a finite number of daughter cells before arresting (*Mortimer and Johnston, 1959*). Because replicative aging is asymmetric, the process of aging occurs in mother cells but is absent in daughter cells (*Egilmez and Jazwinski, 1989*; *Kennedy et al., 1994*). Several asymmetric phenotypes have been identified and proposed to contribute to mother cell decline (*Sinclair and Guarente, 1997*; *Lai et al., 2002*; *Aguilaniu et al., 2003*; *Erjavec et al., 2007*; *Eldakak et al., 2010*; *McFaline-Figueroa et al., 2011*; *Nystrom and Liu, 2014*). We recently found that the acidity of the yeast lysosome-like vacuole is asymmetric between mother and daughter cells. Vacuole acidity declines in mother cells in early age and limits lifespan, but daughter cells have acidic vacuoles (*Hughes and Gottschling, 2012*). To identify what reduces vacuole acidity and how vacuole acidity is regenerated in daughter cells, we further characterized vacuole pH asymmetry. Cells were aged using a genetic system (*Lindstrom and Gottschling, 2009*) and vacuole acidity was monitored by staining cells with quinacrine, a fluorescent dye that accumulates in the acidic vacuole (*Weisman et al., 1987*). We observed bright vacuolar quinacrine staining indicative of acidic pH in a high percentage of buds (nascent daughter cells) regardless of mother cell age, whereas staining was diminished or undetectable in mother cell vacuoles (*Hughes and Gottschling, 2012*) (*Figure 1A*). Thus, throughout their lifespan mother cells produce daughter cells capable of regenerating vacuole acidity.

To further characterize vacuole pH asymmetry, the timing of re-acidification of the bud vacuole was examined. Vacuole acidity was asymmetric in cells treated with nocodazole (*Figure 1B*), suggesting that the bud vacuole re-acidifies prior to cytokinesis. We confirmed that re-acidification occurred before cytokinesis by examining cells containing the septin marker Cdc10-mCherry (*Figure 1C*). A single septin ring at the bud neck transitions to two rings during cytokinesis (*Lippincott et al., 2001*). We observed high vacuole acidity in buds when there was a single septin ring, further supporting that vacuole acidity regenerates prior to cytokinesis. Thus, throughout their lifespan, mother cells produce daughter cells that regenerate vacuole acidity prior to cytokinesis, when mother and daughter cells share a common cytosol. Thus, whatever causes the asymmetry of vacuole pH must also be asymmetric between mother and daughter cells prior to cytokinesis.

**\*For correspondence:**
dgottsch@fhcrc.org

**Present address:** [†]Department of Biochemistry, University of Utah, Salt Lake City, United States

**Competing interests:** The authors declare that no competing interests exist.

**Reviewing editor**: Karsten Weis, ETH Zürich, Switzerland

**eLife digest** Aging is a part of life—but its biological basis and, in particular, how aged cells give rise to young offspring (or progeny) has not been clearly established for any organism.

Budding yeast is a microorganism that is a valuable model to understand aging in more complex organisms like humans. Budding yeast cells undergo a process called 'replicative aging', meaning that each yeast mother cell produces a set number of daughter cells in her lifetime. However, when daughter cells arise from an aging mother cell, the daughter's age is 'reset to zero'. How mother cells age, and how their daughters are rejuvenated, are questions that have been studied for decades.

Previously, researchers discovered that a mother cell's vacuole (an acidic compartment that stores important molecules that can become toxic) becomes less acidic as the mother cell ages. Daughter cells, on the other hand, have very acidic vacuoles, which was linked to their renewed lifespans. However, the mechanism behind this difference in the acidity of the vacuole between mother and daughter cells was unknown.

Now, Henderson et al. have found that a protein (called Pma1), which is found at the cell surface and pumps protons out of the cell, is present in mother cells but not in their newly-formed daughter cells. Furthermore, the Pma1 protein also accumulates as mother cells age. By pumping protons out of the cell, Pma1 effectively reduces the number of protons available to acidify the vacuole in the mother cell. However, because at first the daughter does not have Pma1, there are still plenty of protons inside the cell to acidify the vacuole.

When Henderson et al. reduced the activity of Pma1 in mother cells, the entire cell became more acidic, and so did their vacuoles. Conversely daughter cells engineered to have more Pma1 were less acidic and had less acidic vacuoles than normal.

Henderson et al. next asked whether reducing Pma1 activity to create a more acidic cell, could extend the lifespan of cells, and found that indeed cells with less Pma1 activity lived longer. As such, these findings indicate that an asymmetry in the acidity of the cell—caused by unequal levels of the Pma1 protein—contributes to reducing the lifespan of the mother cell and to rejuvenating the daughter cell. Thus Henderson et al. have identified one of the earliest events in the cellular aging process in budding yeast. Their findings suggest that an imbalance in an activity that is normally essential for cell survival (in this case, the activity of Pma1) can have long-term consequences for the cell that lead to aging.

One of the major points of regulation of vacuole acidity is assembly of the vacuolar proton ATPase (V-ATPase), a multi-subunit complex that pumps protons from the cytosol into the vacuole. The V-ATPase consists of the integral membrane $V_0$ complex and the $V_1$ complex that associates with the $V_0$ (*Li and Kane, 2009*). We examined whether there was a difference in vacuole-associated $V_1$ or $V_0$ between mother and daughter cells by visualizing green fluorescent protein (GFP) tagged subunits of each domain and found no evidence of asymmetry (*Figure 1—figure supplement 1*). Thus, no obvious difference in V-ATPase assembly can account for vacuole pH asymmetry between mother and bud.

In a screen to identify proteins asymmetrically retained in mother cells throughout aging, we identified the plasma membrane proton ATPase, Pma1 (*Thayer et al., 2014*). Pma1 is the major regulator of cytosolic pH (*Ferreira et al., 2001*; *Serrano et al., 1986*), and has similar activity to the V-ATPase, in that they both translocate cytosolic protons across membranes. Pma1 pumps protons from the cytosol out of the cell, whereas the V-ATPase pumps cytosolic protons into the vacuole (*Orij et al., 2011*). Because Pma1 regulates cytosolic pH, we hypothesized that it could antagonize vacuole acidity during aging and underlie vacuole pH asymmetry.

As a first step in testing our hypothesis, we analyzed Pma1 protein localization. There are conflicting reports on Pma1 asymmetry (*Smardon et al., 2013*; *Khmelinskii et al., 2012*; *Malínská et al., 2003*), however we found that Pma1 was asymmetric between mother and daughter cells. Pma1 levels at the plasma membrane were higher in mother cells than daughter cells as indicated by indirect immunofluorescence with antibody to Pma1 (*Figure 2A*). Similarly, Pma1 was more abundant in mother cells than buds when visualized with either Pma1-GFP (*Figure 2A*) or Pma1-mCherry fusion protein (*Figure 2B*).

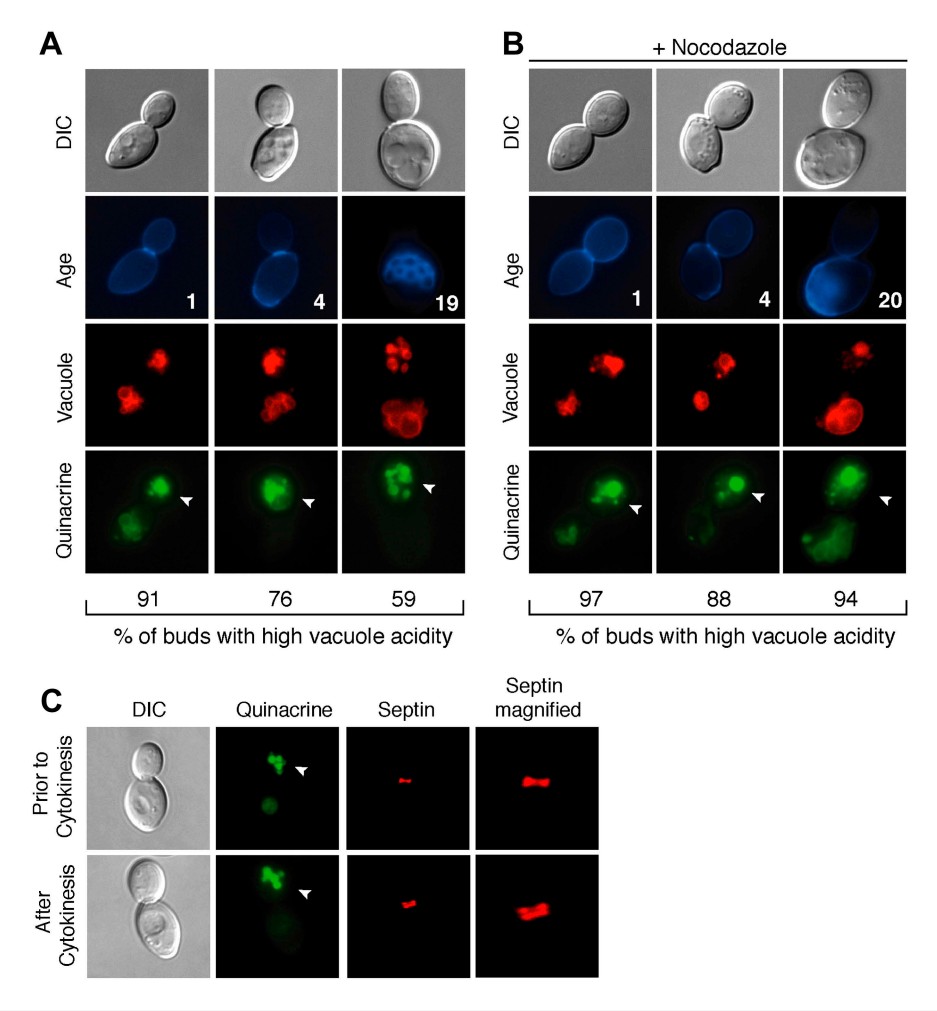

**Figure 1**. Vacuole acidity regenerates in daughter cells throughout mother cell aging and reacidification occurs prior to cytokinesis. (**A** and **B**) Age (# of cell divisions) is shown in the second row and represents exact age determined by calcofluor staining of bud scars. Representative images are shown. n ≥30 cells per timepoint. Arrowheads indicate the daughter cell. DIC, differential interference contrast. (**A**) Vacuole acidity indicated by quinacrine staining of aged cells expressing Vph1-mCherry (vacuole membrane marker). (**B**) Vacuole acidity of cells expressing Vph1-mCherry and arrested prior to cytokinesis by nocodazole treatment. (**C**) Cells with septin morphology indicated by Cdc10-mCherry were quinacrine stained and vacuole acidity was examined before or after cytokinesis (one septin ring or two rings).
The following figure supplement is available for figure 1:

**Figure supplement 1**. Subunits of the V-ATPase are not asymmetric between mother cells and buds.

We also detected mCherry and GFP fluorescence in the vacuole, which likely represents misfolded protein directed to the vacuole for degradation (*Chang and Fink, 1995*). Importantly, we found that asymmetry of Pma1 at the plasma membrane was maintained through at least 18 mother cell divisions and that Pma1 was asymmetric prior to cytokinesis (*Figure 2B*), paralleling the asymmetry of vacuole pH.

When we examined Pma1 distribution during mother cell aging, we found that Pma1 increased at the plasma membrane in early age (*Figure 2B*). Pma1 levels were very low in newborn daughter cells, increased as daughter cells became mothers, and continued to increase over the first three mother cell divisions. This pattern of Pma1 abundance in daughters and aging mother cells inversely correlated with vacuole acidity. When Pma1 was very low in buds and newborn cells, the vacuole was acidic. In contrast, vacuole acidity was reduced in mother cells that have high Pma1 levels.

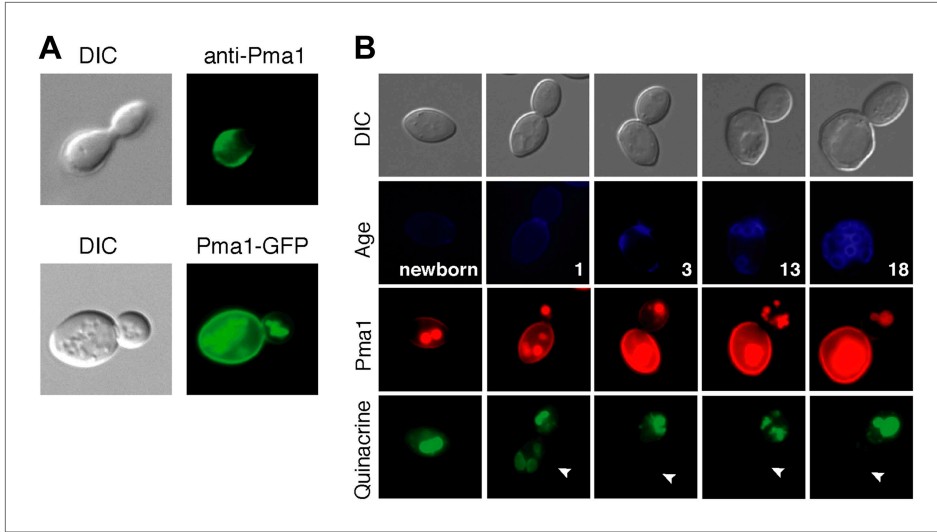

**Figure 2**. Plasma membrane Pma1 levels are asymmetric between mother cells and buds, accumulate with age, and are inversely correlated with vacuole acidity. (**A**) Top panel: Indirect immunofluorescence imaging of Pma1 with anti-Pma1 antibody in untagged young cell. Bottom panel: Pma1-GFP localization in young cell. (**B**) Newborn daughter cells and aged mother cells expressing Pma1-mCherry were quinacrine stained. Arrowheads indicate the vacuoles of interest.

The inverse correlation between Pma1 levels and vacuole acidity suggested that Pma1 could cause vacuole pH asymmetry by antagonizing V-ATPase activity in mother cells. We first tested whether high levels of Pma1 could reduce vacuole acidity by overexpressing an extra copy of *PMA1* in newborn daughter cells from an inducible promoter (*Gao and Pinkham, 2000*; *Veatch et al., 2009*). Overexpression of *PMA1-mCherry* increased Pma1 at the plasma membrane of newborn cells (*Figure 3—figure supplement 1*). Without excess Pma1, 87% of newborn cells had highly acidic vacuoles, whereas vacuole acidity was only high in 13% of cells upon *PMA1* overexpression (*Figure 3A*). To further test whether Pma1 antagonized vacuole acidity, we reduced Pma1 activity and examined vacuole acidity in aging mother cells. *PMA1* is an essential gene and cannot be deleted (*Serrano et al., 1986*), so we reduced its activity by 65% using the *pma1-105* allele that has a mutation in the catalytic domain (*McCusker et al., 1987*; *Perlin et al., 1989*). In contrast to wild-type cells where vacuole acidity was reduced in more than 80% of cells in the third and subsequent mother cell divisions (*Figure 3B*), *pma1-105* cells retained high vacuole acidity after 3 divisions and up to at least 18 divisions (84% and 79% respectively, *Figure 3B*). These results suggest that Pma1 activity antagonizes vacuole acidification and, combined with the expression pattern of Pma1, support the idea that increased Pma1 in aged mother cells causes the reduction of vacuole acidity.

We previously found that delaying the reduction of vacuole acidity during aging by increasing V-ATPase levels extends replicative lifespan (*Hughes and Gottschling, 2012*). Given the evidence presented above that Pma1 levels antagonize vacuolar acidity, we asked whether reduced Pma1 activity also affected lifespan. Indeed, the *pma1-105* allele increased median replicative lifespan by ~30% (*Figure 3C*), comparable to well-characterized lifespan-extending mutations (*Delaney et al., 2011*). The slope of the *pma1-105* lifespan curve is similar to the slope of the wild-type curve. This suggests that instead of influencing the rate of aging throughout lifespan, the *pma1-105* allele delays the onset of the normal aging process. To ascertain whether lifespan extension by the *pma1-105* allele occurred entirely via increased vacuolar acidity, we examined the lifespan of *pma1-105* cells that lacked V-ATPase function. Cells lacking the V-ATPase subunit Vma2 had a short median lifespan of 2 divisions, as previously reported (*Hughes and Gottschling, 2012*). The lifespan of cells that had reduced Pma1 activity and that were devoid of V-ATPase function (*vma2Δ, pma1-105*) was much shorter than wild-type lifespan (median 7 divisions), and more similar to cells lacking V-ATPase function. This suggests that most of the lifespan extension imparted by the *pma1-105* allele requires V-ATPase function, but that the mechanism of lifespan extension is not limited to increased vacuolar acidification.

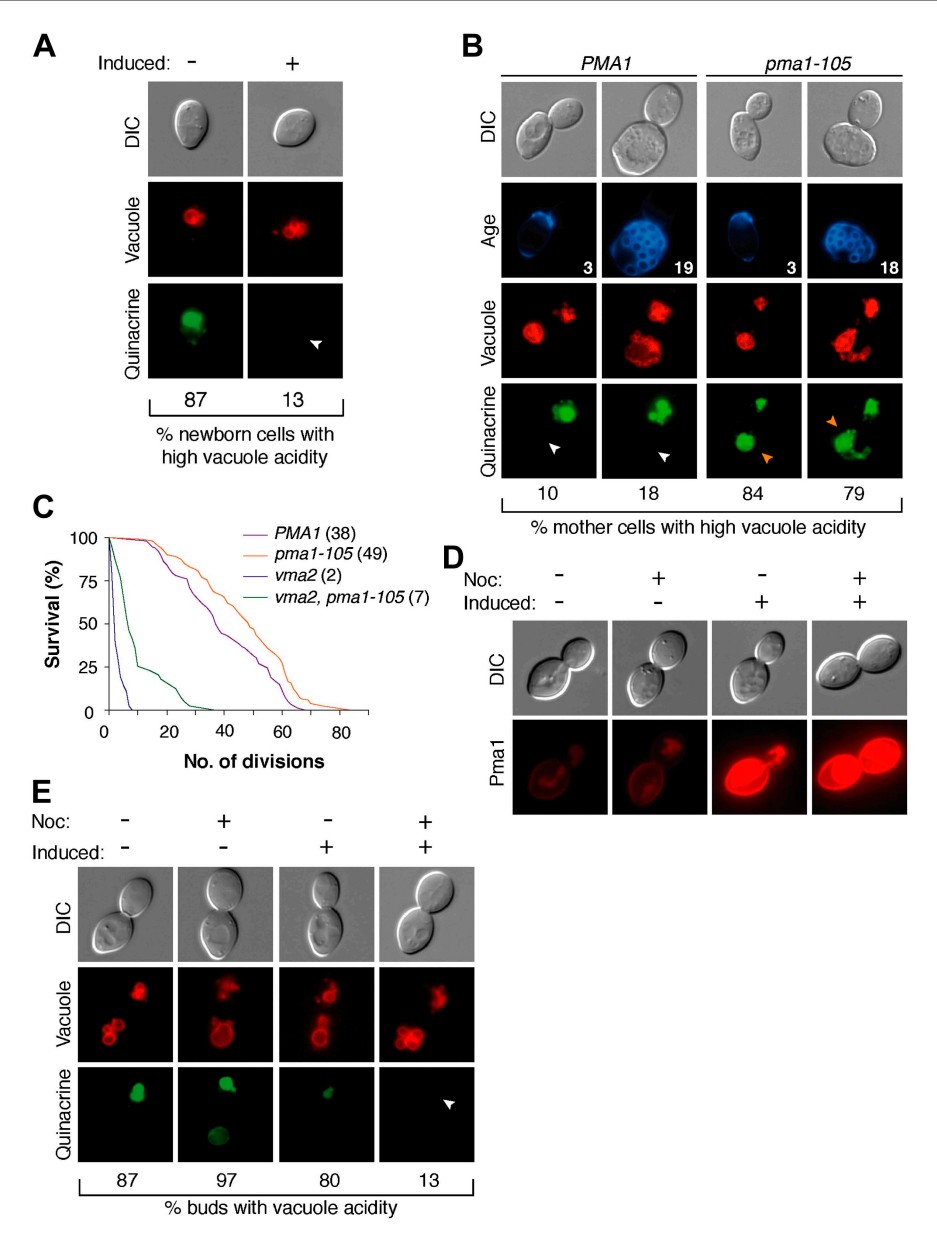

**Figure 3**. Pma1 antagonizes vacuole acidity and its absence facilitates regeneration of vacuole acidity in buds. (**A**) *PMA1* was overexpressed in newborn daughter cells expressing Vph1-mCherry using a β-estradiol inducible system where a GAL4-Estrogen binding domain-VP16 (GEV) fusion protein drives *GAL1* promoter expression of an extra copy of *PMA1*. (n ≥ 30 cells per condition). (**B**) Wild-type and *pma1-105* cells expressing Vph1-mCherry were aged and quinacrine stained (n ≥ 30 cells per timepoint). White arrowheads indicate mother cell vacuoles with reduced acidity. Orange arrowheads indicate acidic mother-cell vacuoles. (**C**) Replicative lifespan of wild-type, *pma1-105*, *vma2*, and *vma2 pma1-105* cells by micromanipulation. Median lifespan is indicated. For the difference between wild-type and *pma1-105*, p < 0.0001, one-tailed logrank test. (n = 114 cells for *PMA1*, n = 119 for *pma1-105*, n = 36 for *vma2*, and n = 39 for *vma2 pma1-105*). (**D**) *PMA1-mCherry* was overexpressed in cells undergoing their first division that expressed endogenous Pma1-mCherry and that were treated with β-estradiol and then with β-estradiol plus nocodazole (Noc). (**E**) As in **D**, cells that expressed Vph1-mCherry were induced to overexpress *PMA1* and were quinacrine stained. (n ≥ 30 cells per condition). Arrowheads indicate the vacuoles of interest.

The following figure supplement is available for figure 3:

**Figure supplement 1**. Overexpression increases Pma1 levels at the plasma membrane.

Taken together these results support the idea that high Pma1 levels on mother cells impair vacuole acidification and limit lifespan.

In addition to Pma1 antagonizing mother cell vacuole acidity with age, we also hypothesized that the inherent asymmetry of Pma1, and thus low levels on buds, allows for re-acidification of the vacuole in buds. To test this idea, we asked whether expressing Pma1 in buds reduced vacuole acidity. We induced overexpression of *PMA1-mCherry* in cells arrested prior to cytokinesis with nocodazole and in untreated cells (*Figure 3D*). In untreated cells, overexpression increased mother cell Pma1 levels but maintained mother-bud asymmetry. However, in nocodazole-arrested cells, *PMA1-mCherry* became equivalently high in mother cells and buds. At least 80% of buds had acidic vacuoles without *PMA1* induction or when *PMA1* was induced in the absence of nocodazole (*Figure 3E*). In contrast, only 13% of buds had acidic vacuoles when Pma1 levels were high in buds. Because increased Pma1 levels in buds impaired re-acidification of the vacuole, we conclude that the inherent asymmetry of Pma1 is required for regeneration of vacuole acidity prior to cytokinesis. We speculate that regeneration of vacuole acidity is required for daughter cell rejuvenation and that if high levels of Pma1 were induced in the buds of aging mother cells, daughter cells would not rejuvenate.

We wondered how Pma1 antagonizes vacuole acidity and how low Pma1 levels in buds permit vacuole reacidification. Given that Pma1 pumps protons out of the cell, we hypothesized that increased Pma1 activity antagonizes vacuole acidity by reducing cytosolic protons available to the V-ATPase. A prediction of this hypothesis is that cytosolic pH may become more basic with age, and may differ between mother and daughter cells. To test this hypothesis, we examined cytosolic pH with ratiometric pHluorin (a pH-sensitive GFP) (*Miesenbock et al., 1998*) fused to a plasma membrane targeting sequence (residues 1–28 of the Psr1 protein) (*Siniossoglou et al., 2000*). With this reagent, cytosolic pH at the cell cortex was visualized (*Figure 4A*) and quantified in mother cells of varying ages and in their buds. Newborn daughter cells or mother cells that had undergone 1 or 2 divisions had a mean cytosolic pH of ~7.1, similar to previous measurements of bulk log phase cultures (*Orij et al., 2009*) (*Figure 4B*). However, as mother cells aged, cortical cytosolic pH increased as much as ~0.5 pH units (*Figure 4B*). Moreover, when we examined mother cells (on average 3 or 18 divisions old) and their attached buds, cortical pH was ~0.2 or ~0.1 pH units lower in buds than mother cells (*Figure 4C*). All together, these results indicate that cortical cytosolic pH increases during replicative aging and is asymmetric between mother and daughter cells. Mother-daughter asymmetry of cytosolic pH might be surprising given the rapid diffusion of protons (*Wraight, 2006*). However, local cytosolic pH differences have been observed in tumor cell invadopodia (*Magalhaes et al., 2011*) and cytosolic pH gradients can form during polarized growth (*Feijo et al., 1999*; *Gibbon and Kropf, 1994*).

To test whether Pma1 asymmetry was required for cortical cytosolic pH asymmetry, we modulated Pma1 levels and activity in mother cells and buds and monitored cytosolic pH. We overexpressed *PMA1*, which increased mother cell levels but maintained mother-bud asymmetry (as in *Figure 3D*), and we overexpressed *PMA1* in nocodazole treated cells to generate equivalent high levels of Pma1 in mother cells and buds. Overexpression of *PMA1* in mother cells alone elevated mother cell cytosolic pH by ~0.7 pH units and increased the mother-bud difference in cytosolic pH by ~0.5 pH units (*Figure 4—figure supplement 1A*). However, when Pma1 levels in buds were elevated to mother cell levels, bud pH increased, abrogating cytosolic pH asymmetry. Moreover, decreasing Pma1 activity with the *pma1-105* allele decreased cortical cytosolic pH by ~0.7 pH units (*Figure 4—figure supplement 1B*). Taken together our results suggest that the inherent asymmetry of Pma1 creates mother-daughter cytosolic pH asymmetry prior to cytokinesis.

Our findings support the idea that Pma1 activity antagonizes vacuole acidification via a competition with the V-ATPase for limited cytosolic protons (at pH 7 there are ~3000 free protons per yeast cell, ~$10^6$ Pma1 molecules and ~$10^5$ V-ATPase $V_0$ subunits) (*Orij et al., 2011*; *Huh et al., 2003*). While other modes of regulation may also be involved, our results can be explained by high Pma1 activity in aged mother cells translocating a sufficient number of protons out of the cytosol to restrict proton availability for the V-ATPase and reduce vacuole acidity (*Figure 4D*). Conversely, lower Pma1 activity in buds leads to more cytosolic protons and higher vacuole acidification.

Our findings identify increased cytosolic pH as an early contributing step to the aging process in budding yeast and suggest that cytosolic pH asymmetry facilitates daughter cell rejuvenation. Interestingly, this same discontinuity of cytosolic pH in plant and algae cells (*Feijo et al., 1999*; *Gibbon and Kropf, 1994*) and of Pma1 orthologs in fission yeast and pollen tubes (*Minc and Chang, 2010*; *Certal*

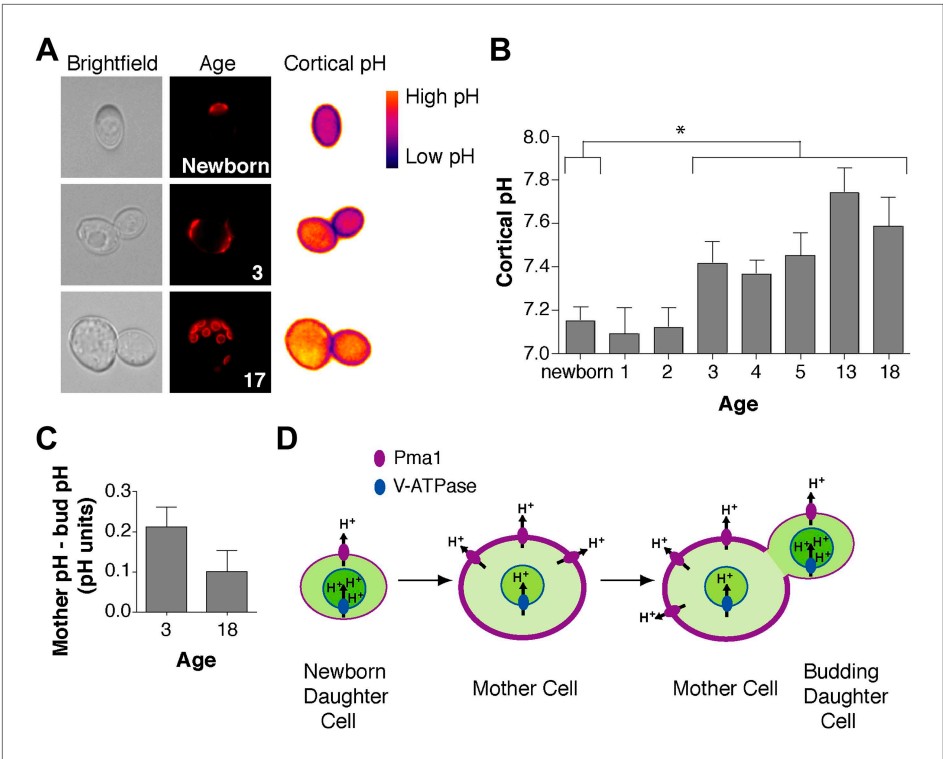

**Figure 4**. Cortex-proximal cytosolic pH increases with age and is asymmetric between mother cells and buds. (**A**) Visualization of cortical pH of newborn and aged mother cells and their buds as indicated with a plasma membrane-anchored ratiometric pHluorin using its bimodal excitation spectrum. Age is indicated by wheatgerm agglutinin-Alexa 594 staining of budscars, which also detects birth scars on newborn cells. (**B**) As in **A**, measurement of cortical pH of newborn and aged mother cells was made at the plasma membrane. n ≥13 cells per timepoint. Mean cortical pH is significantly increased in mother cells undergoing their third division and thereafter compared to newborn cells (p ≤ 0.014, one-tailed unpaired t test). Error bars represent SEM. (**C**) Difference of the cortical pH of mother cells and their buds. Bud pH was lower (more acidic) than mother cell pH (p = 0.003, n = 17 cells at 3 divisions and p = 0.04, n = 16 cells at 18 divisions, one-tailed paired t tests) and was subtracted from mother cell pH. (**D**) Model of the effect of Pma1 asymmetry and increased Pma1 levels during aging on the magnitude of proton translocation out of the cytosol and into the vacuole.

The following figure supplement is available for figure 4:

**Figure supplement 1**. Pma1 asymmetry mediates cortical cytosolic pH asymmetry.

---

et al., 2008) is conserved during polarized cell growth. We speculate that non-uniform cytosolic pH is generally important in polarized growth and has been co-opted in cell types that undergo asymmetric divisions to regenerate full cellular capacity.

## Materials and methods

### Strains

Yeast strains are listed in *Supplementary file 1*. All strains are derivatives of *Saccharomyces cerevisiae* S288c (BY) (*Brachmann et al., 1998*). One-step PCR-mediated gene replacement and epitope tagging were performed using standard techniques, with template plasmids pRS306, pRS400, pKT127 and pKTmCherryKanMX (*Sheff and Thorn, 2004*; *Sikorski and Hieter, 1989*). Oligonucleotides for gene replacement, tagging, and cloning are listed in *Supplementary file 1*.

Strains expressing Vma2–GFP were derived from the yeast GFP collection (*Huh et al., 2003*). The Gal4-Estrogen binding domain-VP16 (GEV) fusion protein (*Veatch et al., 2009*) was integrated into the *leu2Δ0* allele by transforming PmeI-linearized pAGL plasmid. Strains expressing *PMA1* or *PMA1-mCherry* from a *GAL* promoter were constructed by transformation of GEV yeast

strains with NotI-digested pAG306-GAL-PMA1 chr1 or pAG306-GAL-PMA1-mCherry chr1, which integrated them into an empty region of chromosome 1 (199456–199457) (*Hughes and Gottschling, 2012*).

The *pma1-105* mutant was derived from the heterozygous yeast deletion collection strain, *pma1Δ::KanMX4/PMA1* (*Winzeler et al., 1999*). The strain was transformed with a linear fragment (derived from NotI and SacII digestion of pma1-105-URA3 plasmids) containing *pma1-105* marked with URA3. We chose transformants that replaced the *pma1Δ::KanMX4* allele similar to a previously described strategy (*Harris et al., 1994*). This heterozygote was sporulated to obtain *pma1-105* haploids, which were sequenced to verify the mutation. The *PMA1 URA3* variant was created from the *pma1Δ::KanMX4/PMA1* strain; the *ura3Δ0* mutation was converted to *URA3* via PCR amplification of *URA3* from pRS306 and transformation. This resulting diploid strain was sporulated to obtain a *PMA1 URA3* haploid. All *PMA1* or *pma1-105* strains that expressed other markers were created by backcrossing to these original haploids.

Strains carrying the cortical pHluorin were constructed by amplifying plasmid pADH1pr-PSR1-RMP with primers (URA3-tTA-intChr1F and R) that allowed insertion into a second empty region of chromosome 1 (17068–17161).

## Plasmids

pKTmCherryKanMX (a kind gift from W Shou) was obtained by digestion of pKT127 (*Sheff and Thorn, 2004*) with PacI and BglII and insertion of the mCherry containing fragment from similarly digested pBS34. pBS34 was obtained from the Yeast Resource Center at the University of Washington with permission from R. Tsien (*Shaner et al., 2004*).

The GEV plasmid was previously described (*Veatch et al., 2009*). pAG306-GAL-PMA1chr1, was generated in two steps. First, we created pAG306-GAL-ccdBchr1, a plasmid for gene expression from the *GAL* promoter that can be integrated into chromosome 1 (199456–199457) after NotI digestion. We generated pAG306-GAL-ccdBchr1 by ligation of a SmaI-digested fusion PCR product that contained two ~500-base-pair regions of chromosome 1 flanking a NotI site into AatII-digested pAG306-GAL-ccdB (Addgene plasmid 14139) (*Alberti et al., 2007*). We generated the fusion PCR product using oligonucleotides ChrI PartB SmaI F and ChrI PartA SmaI R to amplify two templates generated by PCR of yeast genomic DNA using oligonucleotide pairs ChrI PartA NotI F and ChrI PartA SmaI R, and ChrI PartB SmaI F and ChrI PartB NotI R, respectively. Second, we inserted *PMA1* into pAG306-GAL-ccdBchr1 from donor Gateway plasmid pDONR221-PMA1 (Harvard Institute of Proteomics [HIP] accession ScCD00008895) (*Hu et al., 2007*), using LR Clonase according to the manufacturer's instructions (Invitrogen, Carlsbad, CA).

pAG306-GAL-PMA1-mCherryChr1 was generated by Gibson Assembly according to the manufacturer's instructions (New England Biolabs, Ipswich, MA). First *PMA1-mCherry* was amplified from genomic DNA from strain UCC9645 using primers GibRXNpAGdest_Pma1ChryF and GibRXNpAGdest_Pma1ChryR and assembled with a PCR product amplified from pAG306-GAL-ccdBChr1 with primers GibsonRXNpAGdstnF-2 and R-2.

pADH1pr-PSR1-RMP was derived from pADH1pr-RMP using Quikchange Site-Directed Mutagenesis (Stratagene, La Jolla, CA) to insert the first 28 amino acids of *PSR1* between the *ADH1* promoter and the N-terminus of RMP (ratiometric pHluorin) using primers PSR1-28-RMpHluorinF and R. pADH1pr-RMP was generated in four steps. First, pKT127-RMP was created by removing GFP from pKT127 by restriction digestion with PacI-AscI and replacing it with similarly digested RMP generated by PCR of template plasmid pGM1 (*Miesenbock et al., 1998*) using oligonucleotides SEP PacIF and SEP AscIR. pADH1pr-RMP was created when RMP and the ADH1 terminator were amplified with primers UPGFP/pHluorin F and R from pKT127-RMP, digested with EcoRI-EagI, and ligated into similarly digested backbone of COX4-dsREDURA3int. COX4-dsREDURA3int was created by ligating the XhoI-NotI fragment of pHS12 (*Bevis and Glick, 2002*) containing the ADH1 promoter and COX4 mitochondrial presequence fused to dsRED.T4 into similarly digested pRG919. pRG919 was created when a SacI-SacII PCR fragment containing a *URA3* targeting construct was inserted between the SacI-SacII sites in pRS406 (*Christianson et al., 1992*).

The PMA1-URA plasmid was created in two steps. First, the *URA3* gene was inserted downstream of the previously characterized *PMA1* transcriptional termination sites (*Nagalakshmi et al., 2008*; *Yassour et al., 2009*) on chrVII (479252–479253) with primers MRKdownPma1F and MRKdownPma1R to create UCC9656. Genomic DNA from this strain was amplified with primers Pma1_1kbupNotIF and Pma1_1kbdownSacIIR to acquire a fragment containing the entire *PMA1* locus plus 1 kb upstream and

1 kb downstream and *URA3*. This fragment was digested with NotI and SacII and ligated into similarly digested pBluescript SK+ (Stratagene). The pma1-105-URA plasmid was generated by Quikchange Site-Directed Mutagenesis (Stratagene) of the PMA1-URA plasmid using primers pma1-S368FF and pma1-S368FR.

## Media and cell culture

Cells were cultured in YEPD (1% yeast extract, 2% peptone, 2% glucose) and maintained in exponential growth for 15 hr to a maximum density of $5 \times 10^6$ cells $ml^{-1}$ before initiating experiments. Where indicated, cells were treated with nocodazole (Sigma, St. Louis, MO) at 10 µg $ml^{-1}$ for 1.5 hr or with 5 µM β-estradiol (Sigma) for 2 hr to induce *PMA1* or *PMA1-mCherry* overexpression.

## Culturing and purification of aged MEP cells

Cells were cultured, biotin labeled, aged, and purified for quinacrine staining as previously described (*Hughes and Gottschling, 2012*). For cortical pH analysis of aged cells, cell labeling and purification were performed as described (*Hughes and Gottschling, 2012*) except incubation with streptavidin-coated magnetic beads (MicroMACS, Miltenyi Biotec, Bergisch Gladbach, Germany) and purification took place in YEPD depleted of biotin. This was achieved by overnight incubation at 4°C of 45 ml YEP with 300 µl Avidin-Agarose beads (Sigma). Glucose was added to 2%. Cells were recovered for 1 hr in YEPD prior to imaging.

## Quinacrine staining, indirect immunofluorescence and fluorescent microscopy

Pma1 was detected by indirect immunofluorescence as described (*Burke et al., 2000*) using the 40B7 monoclonal antibody (Abcam, Cambridge, England) followed by Alexa Fluor 488-conjugated goat anti-mouse secondary (Invitrogen).

Quinacrine (Sigma) staining was performed as previously described (*Hughes and Gottschling, 2012*). In most experiments, age was determined by calcofluor (Sigma) staining of bud scars by including 5 µg $ml^{-1}$ calcofluor in the last wash step before imaging. Calcoflour staining reveals bud scars (*Pringle, 1991*) and facilitates identification of newborn cells and the replicative age of mother cells. Calcofluor also stains the mother cell birth scar (*Pringle, 1991*) and allowed identification of the new mother cell during nocodazole arrest when mother cells and buds are similar in size.

Cells were imaged under ×60 oil magnification using a Nikon Eclipse E800 (Nikon, Tokyo, Japan) with the appropriate filter set: UV-2E/C DAPI for calcofluor; FITC-HYQ for quinacrine, GFP and Alexa Fluor 488; and G-2E/C TRITC for mCherry. Images were acquired with a CoolSNAP HQ$^2$ CCD camera (Photometrics, Tucson, AZ) and Metamorph version 7.1.1.0 imaging software (Molecular Devices, Sunnyvale, CA).

## Lifespan measurement by micromanipulation

Replicative lifespan was measured by micromanipulation as previously described (*Hughes and Gottschling, 2012*).

## Single-cell analysis of cortical pH of mother cells and buds

Cells were cultured in YEPD, but transferred to low fluorescence medium (*Orij et al., 2009*) for pH measurement after rinsing them in an equal volume of low fluorescence medium. Calibration curves were created as previously described (*Orij et al., 2009*) except that that cells were permeabilized prior to pH equilibration by treatment in 5 µg $ml^{-1}$ digitonin (Sigma) in 1× PBS for 5 min. To quantify replicative age, $1 \times 10^7$ cells were stained in YEPD for 5 min with 10 µg $ml^{-1}$ Wheat Germ Agglutinin-Alexa fluor 594 conjugate (Molecular Probes, Eugene, OR), washed once with an equal volume of YEPD, once with low fluorescence medium, and transferred to low fluorescence medium for 20 min prior to imaging at a density of $1 \times 10^7$ cells $ml^{-1}$.

To quantify cortical pH of live single cells and to generate pH calibration curves, cells were imaged with a Leica DMI6000 B under ×63 oil magnification. Images were acquired with a Leica DFC365 FX camera and Leica Application Suite Advanced Fluorescence software (Leica, Wetzlar, Germany). TRITC excitation and emission filters (Ex525/25, Em605/52) were used to image bud scars and the combination of FITC Excitation/FITC Emission (Ex490/20, Em525/36) and DAPI Excitation/FITC Emission (Ex402/15, Em525/36) filters were used to derive cortical pH using the bimodal excitation spectrum of ratiometric pHluorin (*Miesenbock et al., 1998*) to calculate 402/490-nm excitation ratios. Image

analysis was performed using ImageJ (Version 1.47m, NIH) to quantify mean pHluorin intensity from the identical regions of images acquired at 402 and 490 nm excitation with a 2 pixel-wide freehand line tool traced along the majority of the length of the mother cell or attached bud plasma membrane. Local background was calculated from a cell-free region one cell diameter away and subtracted from all membrane intensity measurements prior to calculation of 402/490-nm excitation ratios that were fitted to calibration curves to derive cortical pH. Statistical analyses were performed using GraphPad Prism version 4.0a software (GraphPad, La Jolla, CA).

We and others (*Pineda Rodo et al., 2012*) note that repeated imaging differentially affected signal intensity captured at 402 and 490 nm excitation wavelengths, which altered the excitation ratios of sequential images in a pH-independent manner. Therefore we captured a single set of images per cell and never exposed cells to excitation wavelengths prior to pHluorin imaging.

## Acknowledgements

We thank M Patel, J Roberts, and members of the Gottschling laboratory for reviewing the manuscript; A Merz for insightful discussions; G Smits, and R Orij for technical advice; G Miesenbock, J McCusker, and W Shou for reagents; and J Hsu, Z Tan, N Thayer, A Waite, and N Yazvenko for technical assistance.

## Additional information

### Funding

| Funder | Grant reference number | Author |
| --- | --- | --- |
| National Institutes of Health | AG023779 | Daniel E Gottschling |
| Glenn Foundation for Medical Research | | Daniel E Gottschling |
| Helen Hay Whitney Foundation | | Adam L Hughes |
| Leukemia and Lymphoma Society | | Kiersten A Henderson |
| National Institutes of Health | AC965722 | Kiersten A Henderson |
| National Institutes of Health | AG000057 | Adam L Hughes |
| National Institutes of Health | AG043095 | Adam L Hughes |

The funders had no role in study design, data collection and interpretation, or the decision to submit the work for publication.

### Author contributions

KAH, Conception and design, Acquisition of data, Analysis and interpretation of data, Drafting or revising the article; ALH, Drafting or revising the article, Contributed unpublished essential data or reagents; DEG, Conception and design, Analysis and interpretation of data, Drafting or revising the article

### Author ORCIDs

Kiersten A Henderson, http://orcid.org/0000-0002-2295-1232
Adam L Hughes, http://orcid.org/0000-0002-7095-3793
Daniel E Gottschling, http://orcid.org/0000-0002-7303-6552

## Additional files

### Supplementary file

• Supplementary file 1. (**A**) Yeast Strains. (**B**) Oligonucleotides.

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
