## [Decision Letter]

Thank you for sending your work entitled “Mother-Daughter Asymmetry of pH Underlies Aging and Rejuvenation in Yeast” for consideration at *eLife*. Your article has been favorably evaluated by Randy Schekman (Senior editor) and 3 reviewers, one of whom, Karsten Weis, is a member of our Board of Reviewing Editors.

The Reviewing editor and the other reviewers discussed their comments before we reached this decision, and the Reviewing editor has assembled the following comments to help you prepare a revised submission.

In this manuscript, Henderson, Hughes, and Gottschling report on the asymmetric localization of the plasma membrane ATPase Pma1, the major proton pump in the plasma membrane of budding yeast. They show that Pma1 plays a role in establishing the asymmetry in vacuolar pH between yeast mother and daughter cells that was previously identified by the authors. They demonstrate that Pma1 is enriched at the plasma membrane of mother cells and absent from buds. This correlates with an increase in the cortical pH in the mother cell compared to the daughter, and a less acidic vacuole in the mother than in the bud. Upon over-expression of Pma1 in cells arrested in mitosis, the asymmetry of Pma1 localization is lost, and the pH asymmetry between mother and bud are reduced or lost both at the cortex and in the vacuole. Likewise, reducing the activity of Pma1 restores the lower pH in the vacuole of mother cells and extends their lifespan. Thus, the authors conclude that asymmetric distribution of Pma1 contributes to the ageing of mother cells and the formation of young daughter cells.

The referees agreed that the data are interesting and overall of good technical quality. However, there was also a consensus that some additional experiments are needed to solidify the results and the conclusion of the authors:

Major points:

1) There was a concern about the low number of cells that were used in the ageing experiments (both wt and pma1-1) and the experiments seem to have been done only once. Higher numbers will be needed to evaluate the mortality of the cells as a function of age (see also point 3).

2) PMA1 mutants are long-lived and the model indicates that this is because the pH balance is altered as fewer protons are being competed for with V-ATPase to acidify the vacuole. If so, does loss of V-ATPase suppress the long lifespan of pma1 mutant yeast? Does overexpression of V-ATPase result in a long lifespan?

3) Another issue that was discussed by the reviewers concerns the interpretation of the results. Does the higher pH of the mother cell's vacuole contribute directly to ageing, or does it simply create a burden to the cell that impacts longevity? By definition ageing causes mortality (frequency of death at a certain time point) to increase with age. Thus, when the log of mortality is plotted as a function of age, the curve should have a positive linear slope. Furthermore, a process contributing to ageing should increase the slope of the curve. A rapid evaluation of the curves presented in Figure 3 suggests that in the pma1 mutant the curve is only shifted relative to the wild type and that the slope is not affected. Thus, it appears as if the pH of the vacuole modulates the time at which ageing “starts” without affecting that process itself. Furthermore, if low pH resets age in the bud, mutants with high Pma1 activity in the bud should show a shorter lifespan that decreases proportionally to the age of their mothers at the time of cytokinesis (i.e. they inherit age and cannot reset it). Alternatively, the cells might not show any difference to their mothers (everybody restarts with the same slight deficit due to the fact that ageing starts earlier for all). While the reviewers agreed that experiments addressing some of these points are laborious (but doable) these issues need to be at least discussed. Furthermore, an increase in the number of cells evaluated (see point 1) will help to get a better picture of how mortality changes with age.

---

## [Author Response]

Major points:

*1) There was a concern about the low number of cells that were used in the ageing experiments (both wt and pma1-1) and the experiments seem to have been done only once. Higher numbers will be needed to evaluate the mortality of the cells as a function of age (see also point 3)*.

In response to reviewers’ suggestion, we tripled the number of wild-type and *pma1-105* cells whose lifespan we analyzed (now 114 and 119 cells, respectively). The additional data support our previous conclusion and are included in Figure 3.

2) PMA1 mutants are long-lived and the model indicates that this is because the pH balance is altered as fewer protons are being competed for with V-ATPase to acidify the vacuole. If so, does loss of V-ATPase suppress the long lifespan of pma1 mutant yeast?

To address the reviewers’ suggestion, we have added lifespan analysis of strains lacking V-ATPase activity (via deletion of *vma2*) in the wild-type *PMA1* or *pma1-105* background and these data are now included in Figure 3. The median lifespan of the *vma2Δ, pma1-105* strain (7 divisions) is much shorter than wild-type (38 divisions) and more similar to the *vma2Δ* lifespan (2 divisions). Thus, the *pma1-105* allele no longer extends lifespan in the absence of V-ATPase activity, suggesting that the ability to acidify the vacuole is required for the majority of the effect of the *pma1-105* allele on lifespan. However, the *vma2Δ, pma1-105* lifespan is extended (7 divisions) compared to *vma2Δ* lifespan (2 divisions), suggesting that the *pma1-105* allele contributes additional benefits besides increased vacuole acidity. We have also added a discussion of these results to the text.

Does overexpression of V-ATPase result in a long lifespan?

We previously found the overexpression of *VMA1* (a component of the V-ATPase) extends median lifespan by 40% (Hughes and Gottschling, Nature, 2012) and refer to this in the text.

*3) Another issue that was discussed by the reviewers concerns the interpretation of the results. Does the higher pH of the mother cell's vacuole contribute directly to ageing, or does it simply create a burden to the cell that impacts longevity? By definition ageing causes mortality (frequency of death at a certain time point) to increase with age. Thus, when the log of mortality is plotted as a function of age, the curve should have a positive linear slope. Furthermore, a process contributing to ageing should increase the slope of the curve. A rapid evaluation of the curves presented in*
Figure 3
*suggests that in the pma1 mutant the curve is only shifted relative to the wild type and that the slope is not affected. Thus, it appears as if the pH of the vacuole modulates the time at which ageing “starts” without affecting that process itself*.

We agree with the reviewers’ comment that while the median lifespan of the *pma1-105* strain is extended by 30%, the slope of the *pma1-105* lifespan curve is similar to the wild-type curve and instead appears shifted compared to wild-type. We have increased the number of cells evaluated as suggested, and the slope of the wild-type and *pma1-105* lifespan curves remain similar. We agree with the reviewers’ interpretation of the data that the *pma1-105* allele likely delays the initiation of the events that normally occur during aging (when aging starts), rather than directly affecting the later events of aging.

We previously published that when vacuole acidity is increased during aging by *VMA1* overexpression, median lifespan increases by 40% and the slope of the lifespan curve is visibly decreased compared to wild-type (Hughes and Gottschling, 2012). These findings suggest that reduced vacuole acidity contributes to aging throughout lifespan, and that increasing vacuole acidity does not simply delay the onset of the normal events of aging. The work we present here suggests that Pma1 activity influences lifespan by antagonizing vacuole acidity during aging. We therefore suspect that the *pma1-105* allele simply delays the division at which vacuole acidity is reduced and that once vacuole acidity is reduced, the normal events of aging that follow occur at the normal rate. The shape of the *pma1-105* lifespan curve is thus consistent with our conclusion that Pma1 activity initiates aging by reducing vacuole acidity and we have added a discussion of this idea.

*Furthermore, if low pH resets age in the bud, mutants with high Pma1 activity in the bud should show a shorter lifespan that decreases proportionally to the age of their mothers at the time of cytokinesis (i.e. they inherit age and cannot reset it). Alternatively, the cells might not show any difference to their mothers (everybody restarts with the same slight deficit due to the fact that ageing starts earlier for all). While the reviewers agreed that experiments addressing some of these points are laborious (but doable) these issues need to be at least discussed. Furthermore, an increase in the number of cells evaluated (see point 1) will help to get a better picture of how mortality changes with age*.

As suggested by the reviewers, we have added a discussion of the expectation that increasing Pma1 activity in buds should decrease rejuvenation.